# Dataset of antiarch placoderms (the most basal jawed vertebrates) throughout Middle Paleozoic

Zhaohui Pan[1]★, Zhibin Niu[2]★, Zumin Xian[3], Min Zhu[1,4,5]

[1] Key Laboratory of Vertebrate Evolution and Human Origins of Chinese Academy of Sciences, Institute of Vertebrate Paleontology and Paleoanthropology, Chinese Academy of Sciences, Beijing 100044
[2] College of Intelligence and Computing, Tianjin University, Tianjin 300350,
[3] Institute of Palaeontology, Yunnan Key Laboratory of Earth System Science,Yunnan University, Kunming 650500
[4] CAS Center for Excellence in Life and Paleoenvironment, Beijing 100044
[5] University of Chinese Academy of Sciences, Beijing 100049
★These authors contributed equally to this work.

*Correspondence to*: Min Zhu (zhumin@ivpp.ac.cn)

**Abstract.** Antiarch placoderms, the most basal jawed vertebrates, have the potential to enlighten the origin of the last common ancestor of jawed vertebrates. Quantitative study based on credible data is more convincing than qualitative study. To reveal the antiarch distribution in space and time, we created a comprehensive structured dataset of antiarchs comprising 64 genera and 6025 records. This dataset, which includes associated chronological and geographic information, has been digitalized from academic publications manually into the DeepBone database as a dateset. We implemented the paleogeographic map marker to visualize the biogeography of antiarchs. The comprehensive data of Antiarcha allow us to generate its biodiversity and variation rate changes throughout its duration. Structured data of antiarchs has tremendous research potential, including testing hypotheses in the fields of the biodiversity changes, distribution, differentiation, population and community composition. Also, it will be easily accessible by the other tools to generate new understanding on the evolution of early vertebrates. The data file described in this paper is available on https://doi.org/10.5281/zenodo.5639529 (Pan and Zhu, 2021).

## 1 Introduction

Placodermi is an extinct grade of jawed vertebrates that first occurred in the Silurian and then dominated the Devonian (Carr, 1995; Young, 2010). Recent prevailing phylogenetic hypotheses placed Placodermi as jawed stem-Gnathostomata that is sister to crown-Gnathostomata or modern jawed vertebrates (Brazeau, 2009; Davis et al., 2012; Dupret et al., 2014; Giles et al., 2015; King, 2021; Long et al., 2015;



30    Qiao et al., 2016; Trinajstic et al., 2015; Zhu et al., 2013, 2016). In this scenario, Antiarcha has usually been placed at the most basal position in the Placodermi (Fig. 1), representing the most basal jawed vertebrates. The geospatial-temporal distribution of Antiarcha will thus help us understand the origin and early evolution of jawed vertebrates. Moreover, Antiarcha was a widely distributed, diverse, and successful group within Placodermi from the late Silurian to the end of Devonian (Denison, 1978; Janvier,

35    1996; Zhao et al., 2016; Zhu, 1996) to allow the inference of biogeographic assumptions (Ritchie et al., 1992; Young, 1984b). As a successful vertebrate group during the Devonian, Antiarcha has much contributed to the Devonian stratigraphic correlation. For instance, the biozonation of East Baltic and southern East Antarctica Devonian succession is partly based upon the antiarchs *Bothriolepis*, *Asterolepis*, and *Pambulaspis* (Young, 1974, 1988). Lukševičs (1996) identified 14 bothriolepid species

40    (12 *Bothriolepis* and 2 *Grossilepis*) in the Frasnian-Famennian formations of the East European Platform, proposed nine antiarch assemblages, and set up the most detailed zonation of the Main Devonian Field. However, Blieck et al. (2000) reminded that *Asterolepis*, *Bothriolepis*, and *Remigolepis*, which have different stratigraphic distributions on different continents, must be used with caution for long-distance correlation.

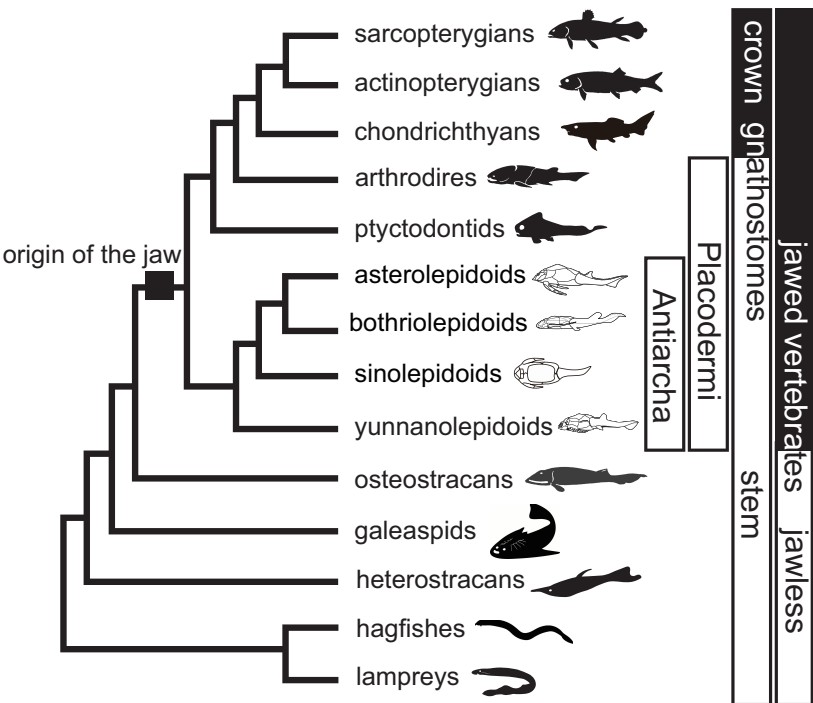

**Figure 1 Phylogenetic relationships of major early vertebrate groups from Qiao et al. (2016) and Pan et al. (2018). Line drawings indicate groups of the Antiarcha. Silhouettes indicate the other vertebrate groups. The black cube on the cladogram points to the orign of the jaw.**

Explaining the spatial and temporal distribution of early vertebrates is the prerequisite to understand their biogeographic exchange. Although Zigaite and Blieck (2013) advocated a quantitative analysis to define biogeographic patterns of early vertebrates, there is still lacking efficient quantitative analysis to understand the dispersal of early vertebrates. This occurs mostly because no comprehensive data collection of early vertebrates was accomplished.

The main objective of this study is to present an unprecedented structured dataset of Antiarcha that is potentially facilitate the research of the geospatial distribution patterns for antiarchs and the quantitive study on early vertebrates. This dataset is the first step to accomplish the global coverage of the vertebrate fossil dataset to analyze the Middle Paleozoic biogeography and paleogeography. Revealing the distribution of antiarchs in the paleomap background with reference to the global paleogeographic reconstructions of Scotese (2002), our preliminary results can be used to test the hypothesis of paleogeographic reconstructions.



## 2. Method

### 2.1 Overview

To facilitate the quantitative study and simulation analysis on early vertebrates, we are looking forward to a comprehensive database. Sallan et al. (2018) pointed out that a lack of early vertebrate

fossil data has limited quantitative approaches, and hindered the resolution of issues in vertebrate evolution. With the implementation of a project entitled "Big Earth Data Science Engineering (CASEarth)" in the Strategic Priority Research Program since 2018 (Guo, 2017), we build the DeepBone database. Via three years of efforts conscientiously, the data of DeepBone database already has reached a certain scale. With continuously refining data, the Antiarcha dataset in the

DeepBone database is the most comprehensive dataset endorsed by Chinese researchers in the Institute of Vertebrate Paleontology and Paleoanthropology, Chinese Academy of Sciences. The Antiarcha dataset differs from the other datasets in its basic unit, which is the specimen ID coupled with the occurrence and other detailed data. All the specimens are referenced to taxa and literature. Because the Antiarcha dataset was designed as a vertebrate paleontological dataset and its input

format was designed as specimen-based, data entry assistants input the metadata according to the published specimen or virtual specimen.

Since there is no satisfactory approach that could automaticly extract the paleontological data from literature, we have recruited several data entry assistants including relevant researchers, master's and PhD students to collect and curate the data. In order to guarantee the quality of the data, we have designed

a four-step data processing procedure (Fig. 2):

1. Data source by an experienced expert who got his PhD degree on early vertebrates.

2. The data entry assistants read the related references, extract the antiarch placoderm data and fill them into the record file under the supervision of experts.

3. Experts review and clean the data according to the references to implement quality control.

4. Further data examinations by senior experts to guarantee the data quality.

5. Data visualization and open access. This step is to provide better user interface to help disseminate the dataset.

Next, we provide more details of the data processing and visualizaiton.


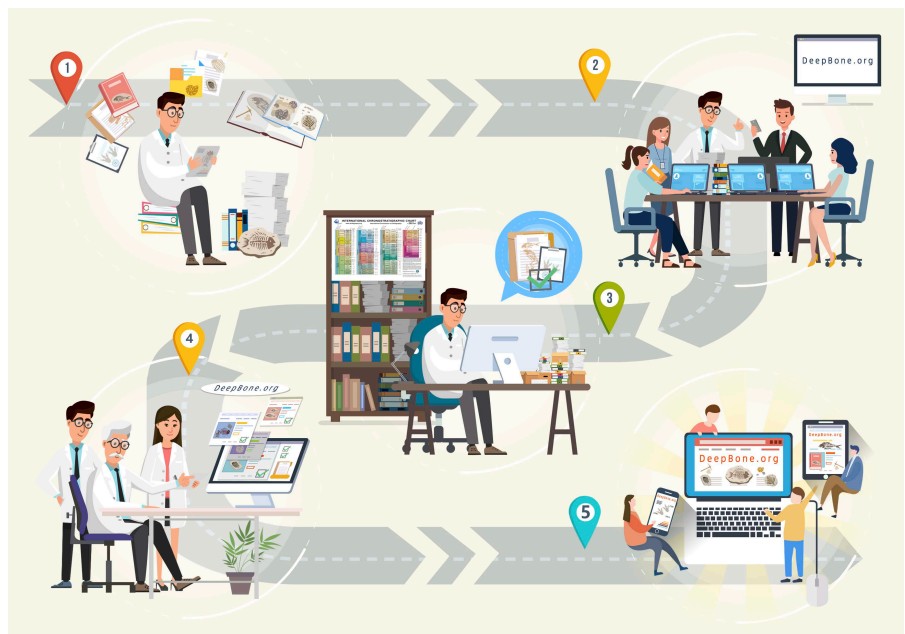

**Figure 2 Workflow of the data processing. Even the web data entry system could be accessed by any registered users, we have recruited several data entry assistants to enrich the data following the standardized workflow. 1, collecting and sifting lecture by experienced experts. 2, data entry assistants digitalize paleontological**

**descriptions from the page into the DeepBone database. 3, experts who are accountable for data review and clean the data according to the references to implement quality control. 4, associated senior researchers review the data. 5, data managers release the data to the public**

## 2.2 Data Source

We invite researchers who have a good reputation in the research scope to sift published literature from

the bibliography of the paleontology journals and books. The main journals includes Alcheringa, Acta Geologica Polonica, Bulletin of the Geological Society of China, Estonian Journal of Earth Sciences, Journal of Vertebrate Paleontology, Journal of Paleontology, Palaeontologia Electronica, Palaeontology, Palaeoworld, and Vertebrata PalAsiatica. Totally, we have collected 126 publications, which span from the year 1939 to the year 2021. The satisfactory literature should include an accurate description or

revision on specimen and taxon. We accepted the latest peer-reviewed literature to deal with the inconsistent descriptions.





### 2.3 Data Processing and Quality Control

We have made a tailored web page that provides a better user interface for them to fill in the rows of paleontological data. After that, the data would be reviewed by the other related experts so that a

researcher could quickly access them to perform quantitative analysis reliably (Fig. 2). This workflow was learned from the Geobiodiversity Database (GBDB) (Xu et al., 2021). Almost all antiarch literature was published in English, Russian, French, German, and Chinese. Data entry assistancestructure could handle the literature in Chinese and English well. Literature in French, Russian, and German was dealt with by paleontology postgraduates who know well these languages.

### 2.4 Data Visualization

The analysis is conducted on the raster tile form paleomaps (Scotese, 1998, 2016). We first convert the excavation locations from current GPS to paleo-GPS using TrackPoint software (Ke et al., 2016; Scotese, 2002). However, the construction of paleontological maps and networks for visual analysis is based on paleomaps. Thus, we convert these latitude and longitude coordinates into pixel coordinates of raster tile

maps using the Web Mercator algorithm. The Web Mercator algorithm is a variant of the Mercator projection and the de facto standard for Web mapping applications (Battersby et al., 2014). Specifically, the projection can be modelled as:

$$\begin{cases} x = a * \theta \\ y = a * \ln \tan(\frac{\pi}{4} + \frac{\varphi}{2}) \end{cases}$$

Where $a$ is the long axis of the earth; $\theta$ is the longitude in radians, the value range is [-π, π], the east

longitude is positive and the west longitude is negative; $\varphi$ is the latitude, the value range is [-π/2, π/2], north latitude is positive and south latitude is negative.

As for the paleomap, we also need to consider the map scaling and canvas size, so the final formula used is as follows:

$$\begin{cases} x_i = l * \theta * 2^{n-1} \\ y_i = l * \ln \tan\left(\frac{\pi}{4} + \frac{\varphi}{2}\right) * 2^{n-1} + m * (i - 1) \end{cases}$$

Where $l$ is the length of the plane map; $n$ is the size of the zoom scale, $n \in N^*$; in the analysis of geological periods, $i$ is taken as 1; in the analysis of cross-geological periods, $i$ represents the $i$-th geological period, $(x_i, y_i)$ are the pixel coordinates of the node in the $i$-th geological period, and $m$ is the fixed pixel distance between the map centers of adjacent geological periods.

### 2.5 Biodiversity Visualization

We calculate the genus and species biodiversity and plot it as a bar map. To better view the results, we

utilize the kernel density estimation (KDE) algorithm to smooth the curve and estimate the biodiversity.

In statistics, KDE is a fundamental data smoothing algorithm that can help inferences about the

population based on finite data samples.

The counts of genus or species of various time slots (visualized as the bins in the charts) are defined as

$(x_1, x_2, \cdots, x_n)$. In statistics, they can be defined as samples drawn from a univariate distribution with an

unknown density $f$. To estimate the shape of this distribution $f$, its kernel density estimator is defined

as

$$\widehat{f_h}(x) = \frac{1}{n}\sum_{i=1}^{n} K_h(x - x_i) = \frac{1}{nh}\sum_{i=1}^{n} K(\frac{x - x_i}{n})$$

where $K$ is the kernel function for estimation, we use the Epanechnikov kernel for better smoothing;

$h > 0$ is a smoothing parameter named as bandwidth. For better assessment of the biodiversity changes,

we adopt the first derivative of the Continuous Probability Distributions function to estimate the rate of

variation.

### 3. Results

### 3.1 Data Overview

This dataset, which was extracted from 126 published papers or books manually, consists of 64 genera

and 6025 records, covering all antiarch lineages. The 6025 records include 5867 fossil specimens that

had been systematically described and documented, and 158 virtual specimens, which were introduced

to describe the taxon information when no specimen was assigned for the referred records. The quantities

distribution of specimens are given in the supplement. Each record has at least one reference within our

dataset, and the specimens lacking precise age are excluded. We transferred the unstructured data from

literature to structured data for further research as detailed as possible. Table 1 shows the framework of

our dataset. Among all the referred specimens, 6.51% belong to Yunnanolepidoidei, 2.86% belong to

Sinolepidoidei, 78.92% belong to 'Bothriolepidoidei', and 11.71% belong to Asterolepidoidei. All the

fossil sites of the constituent groups are plotted in Figure 3 and the phylogeny of Antiarcha follows that

in Pan et al. (2018).



| Specimen record | |
|---|---|
| Formation | Specimen ID |
| Member | Genus |
| Fossil Locality | Species |
| Discovery country | Custodian Institute |
| Latitude | System / Period |
| Longitude | Series / Epoch |
| Paleo-latitude | Stage / Age |
| Paleo-longitude | Group |

| Reference |
|---|
| Literature type |
| Title |
| Keywords |
| Volume |
| Issue |
| Pages |
| Authors |
| Year |
| Journal |
| Doi |

**Table 1 The structure of Antiarcha dataset.**

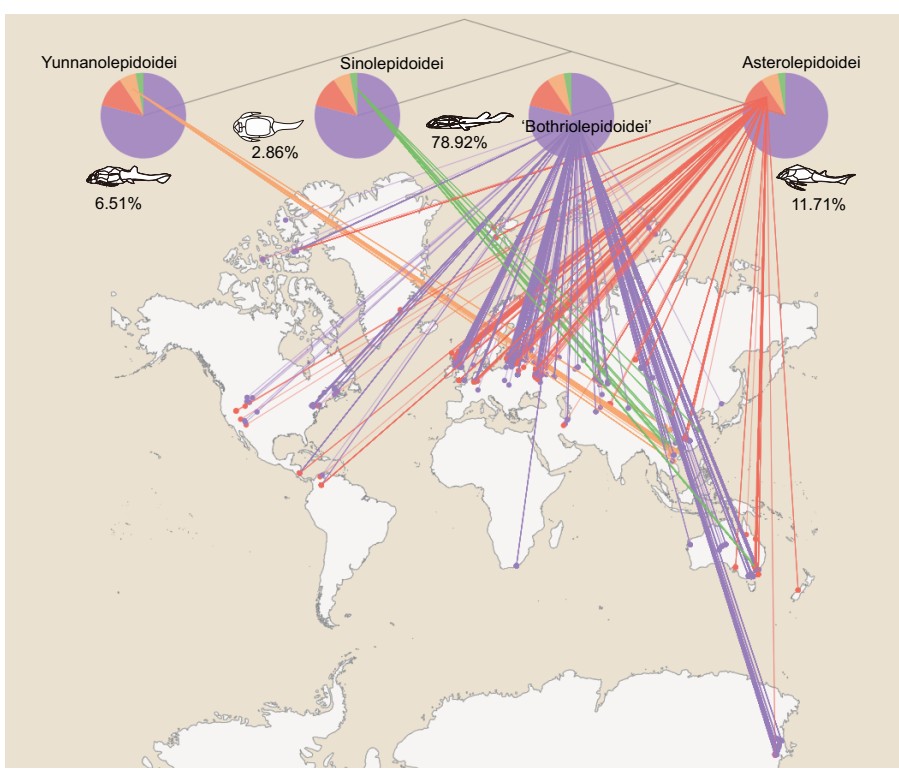

**Figure 3 Graphic abstract of Antiarcha data. The tree topology is simplified from the phylogenetic result in Pan et al. (2018). Numbers under the pie chart represent the relative amounts of each group. Terminal groups are linked with their geographic distributions.**




No antiarch record was in GBDB. Compared to the 138 records of Antiarcha in the Paleobiology Database (PBDB, 2021-08-12), this is the most comprehensive dataset of Antiarcha up to now (Table 2). Moreover, only taxon rank, reference, and occurrence location are available in PBDB. More structured

information of Antiarcha is available in DeepBone. Every record of DeepBone was assigned to an age with the lastest reference. It is open accessable through the website of the DeepBone database or https://doi.org/10.5281/zenodo.5639529.

| | DeepBone Database | Paleobiology Database(PBDB) |
|---|---|---|
| Type | specimen-based | fossil-occurrence-based |
| No. of references | 355 | 19 |
| No. of genera | 64 | 26 |
| No. of species | 187 | 98 |
| No. of Specimens/ occurrences | 6025 | 138 |
| Found in | 2018 | 1998 |
| Websit (last access: 22 October 2021) | www.deepbone.org | https://paleobiodb.org/#/ |

**Table 2 The comparison of data of Antiarcha in two paleontological databases.**

**3.2 The Geospatial Distribution of the Antiarcha Dataset**

As for the fossil site distribution, Yunnanolepidoidei is endemic in the South China block (comprising southern China and northern Vietnam). Sinolepidoidei is limited in South China and Australia (East Gondwana). In contrast, 'Bothriolepidoidei' and Asterolepidoidei are cosmopolitan, especially *Bothriolepis*. The faunistic elements in the communities are used herein at the genus level for their

distributions because many *Bothriolepis* species were described based on isolated plates lacking diagnostic characters (Blieck and Janvier, 1993; Downs, 2011). The heat map of fossil records (Fig. 4) shows that Europe, Australia, and China account for the most records in the world, partly due to their long research history.

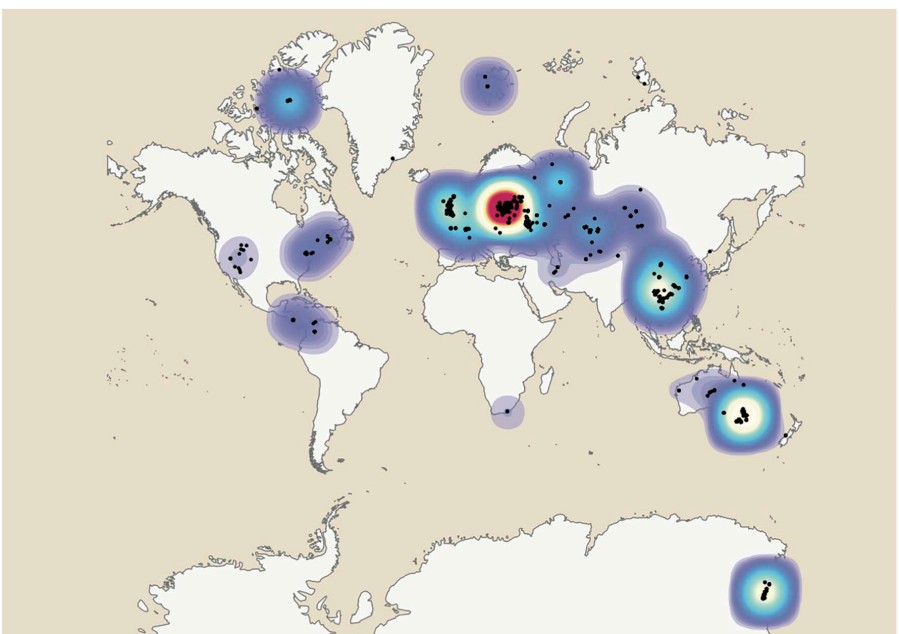

**Figure 4 Heat map of Antiarcha records.**

As Young (1990a) mentioned that biogeographic data must be interpreted in the context of paleogeographic hypotheses, we plot our data on paleomap(Fig. 5)to generate an outline of their past. The analysis of biogeographic trends in Paleozoic vertebrates is highly dependent on fossil data and paleocontinent reconstruction. Due to the easy access of the paleo-geographic coordinates calculator

(PointTrack version 7.0) (Ke et al., 2016; Scotese, 2002), we decided to use Scotese's paleocontinent reconstruction to perform the plot map, although many paleogeographic reconstructions were proposed (Heckel and Witzke, 1979; Li and Powell, 2001; Scotese et al., 1985). The continental reconstructions of Scotese place Baltica,  China and Australia in the tropic and subtropic near the equator from Llandovery to Famennian. The timescale follows the International Commission on Stratigraphy

International Chronostratigraphic Chart version 2021/07.

### 3.3 The Paleogeographic Distribution of the Antiarcha Dataset

We plotted these fossil sites on the paleogeographic map (Fig. 5) except the Silurian *Shimenolepis*, which is the earliest record of Yunnanolepidoidei and the only documented antiarch specimen before the Devonian (Wang, 1991; Zhao et al., 2016). Most of the fossil sites were positioned around the equator.

In the present scenario, the suborder Yunnanolepidoidei apparently originated as early as Silurian in the

South China block, forming a highly endemic fauna. All fossil sites of Yunnanolepidoidei lied in southern China and northern Vietnam (Wang et al., 2010). From Ludlow (Silurian) to Early Devonian, Yunnanolepidoidei formed dominant antiarchs. Sinolepidoidei and 'Bothriolepidoidei' first appeared in Pragian in South China, and Asterolepidoidei first evolved in Emsian in Australia or East Gondwana.

During Middle Devonian, along with lessened isolation of South China, Yunnanolepidoidei became extinct. Euantiarcha ('Bothriolepidoidei' + Asterolepidoidei) dominated the antiarchs in Middle and Late Devonian, and only a few members of Sinolepidoidei coexisted with them in China and Australia. In Eifelian, Asterolepidoidei suddenly bloomed in Baltica without any clue from the older horizons. The distribution and diversity of Antiarcha reached a peak in Givetian. 'Bothriolepidoidei' and

Asterolepidoidei represent the main groups of Antiarcha in Givetian, comprising 5 bothriolepidoid genera with 42 fossil locations and 9 asterolepidoid genera with 49 fossil locations. The records of antiarchs decreased in the Pan-Cathaysian and East Gondwana in Late Devonian, contrary to those increased in Baltica. Antiarchs in Baltica went through an independent evolution in Late Devonian.

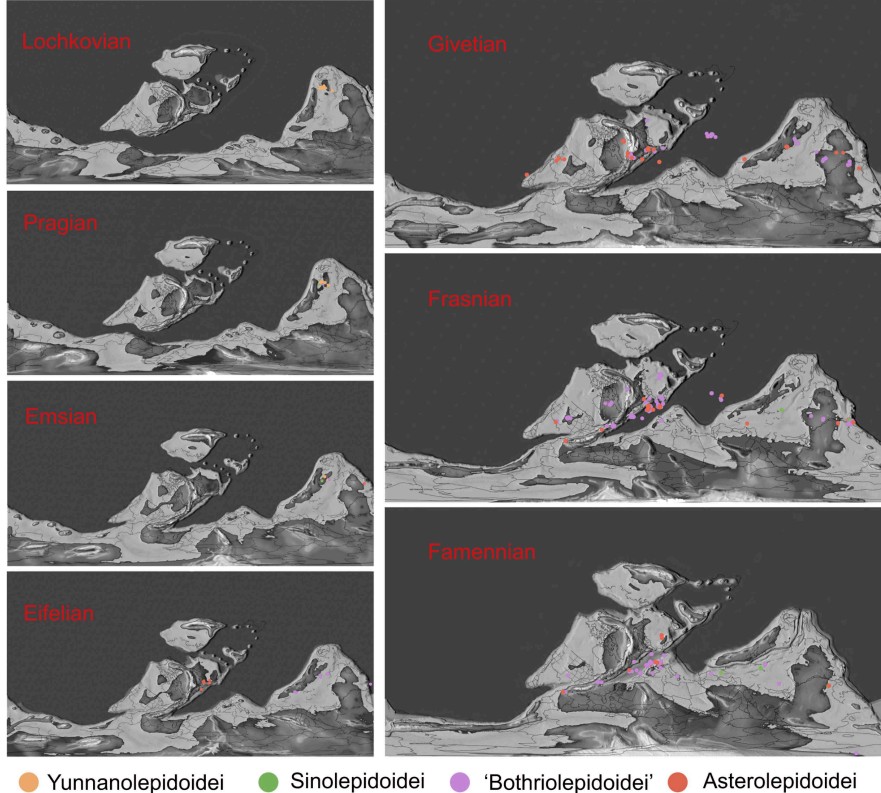



**Figure 5 The distributions of Antiarcha during Devonian. The paleo-coordinates are calculated by TrackPoint. Colors denoting respective groups follow Figure 3.**

## 4 Discussion

### 4.1 First Appearance Record

First appearance record of a taxon or a lineage is important in paleontology and evolutionary biology as
it renders a hard minimum constrain on molecular clock calibration for a taxon (Benton and Donoghue, 2007; Benton et al., 2009; Donoghue and Benton, 2007). Based on our dataset, the oldest record of yunnanolepidoids or antiarchs is *Shimenolepis graniferus* from the Xiaoxi Formation at Shanmen Reservoir, Lixian County, Hunan, China. *Shimenolepis* was first described as the oldest known placoderm, dated as Telychian of Llandovery (Janvier, 1996; Wang, 1991). However, after a detailed
stratigraphic work, Zhao et al. (2016) suggested that the age of *Shimenolepis* is late Ludlow rather than late Llandovery. Janvier and Tông-Dzuy (1998) also documented an indeterminate yunnanolepidoid (Antiarcha gen. sp. indet.) from the Do Son Formation of northern Vietnam, which could be another earliest antiarch potentially.

The oldest sinolepid is *Liujiangolepis suni*, from the Nakaoling Formation (Pragian), Xiangzhou,
Guangxi, China (Wang, 1987). The oldest bothriolepidid is *Houershanaspis zhangi*, documented from the Danlin Formation (Pragian) in Mt. Houershan, Guizhou, southwestern China, on the basis of a bothriolepid-like anterior median dorsal plate (Lu et al., 2017). The earliest asterolepidoid records are represented by *Wurungulepis* and some disarticulated specimens, which had been documented from the Broken River Formation, Broken River, Australia. The age of the Broken River Formation was first
referred to Eifelian, then reassigned to Emsian (*serotinus* Zone) (Burrow, 1996; De Pomeroy, 1996; Young, 1984a, 1990b).

### 4.2 The Origin and Differentiation of Euantiarcha

*Houershanaspis*, *Wudinolepis*, and *Wufengshania* from Pragian and Emsian represent the earliest members of 'Bothriolepidoidei' (Chang, 1965; Lu et al., 2017; Pan et al., 2018), highlighting the origin
and early diversification of this group in South China. In Eifelian, the distribution range of 'Bothriolepidoidei' extended from South China to Iran that was positioned at the northern margin of East Gondwana with the discovery of *Hyrcanaspis* (Janvier amd Pan, 1982). 'Bothriolepidoidei' became cosmopolitan during Givetian and Late Devonian. The earliest asterolepidoid records are from Australia



or East Gondwana in Emsian (Burrow, 1996; Young, 1984a, 1990b). In Eifelian, asterolepidoids
occurred in Baltica, in addition to East Gondwana. Compared to the abundant Early and Middle Devonian
fish faunas of in South China, asterolepidoids were late arrivals to South China (Young, 1984a) since no
asterolepidoid was known from this province before Givetian. It could be inferred that Asterolepidoidei
originated and differentiated in East Gondwana.

**4.3 The Diversity Changes of Antiarcha**

The comprehensive data of Antiarcha allow us to generate its diversity changes through its life span. We
used Kernel Density Estimation to calculate the curves of its biodiversity and variation rate (Fig. 6). The
curves of biodiversity at genus and species levels were similar except for their peaks. The curve of genus-
level reached its peak at Givetian, but the curve of species-level reached its peak at Frasnian. Both curves
could identify two sharp risings in diversity in Pridoli and Eifelian. The first rise in Pridoli coincided
with the rising of the jawed vertebrate and resulted in Yunnanolepidoidei reaching its diversity peak in
Lochkovian. The velocity rate of the genus level reached its peak at the end of Pridoli. After the E'Em
Event, the second rising was recognized by a faunal turnover from endemics to cosmopolitans (Zhu,
2000). Before this event, Petalichtyida and Antiarcha were endemic to China (Pan et al., 2015).
*Wurungulepis* was the only documented antiarch outside China in Early Devonian (Young, 1990b).
Yunnanolepidoidei was extinct after the E'Em Event, coinciding with the differentiation of euantiarchs.
The variation rate of species-level reached its peak at the end of Eifelian. Zhu (2000) has proposed the
Late Eifelian Event (Walliser, 1996) for the Chinese placoderm extinction. Although decreasing in Pan-
Cathaysian, the genus and species diversity of Antiarcha in other parts of the world kept rising after the
Late Eifelian Event. Walliser (1995) proposed the Mid-Famennian Event is characterized by the increase
of the species diversity of Antiarcha. On the contrary, our data shows that the species diversity of
Antiarcha underwent the sharpest decrease from Frasnian to Famennian except for the end of the
Devonian. Only the genus diversity of Antiarcha increased slightly at the beginning of Famennian.
Another difference between the two biodiversity curves is that the decreased genus diversity was in
marked contrast to the increased species diversity from Givetian to Frasnian.

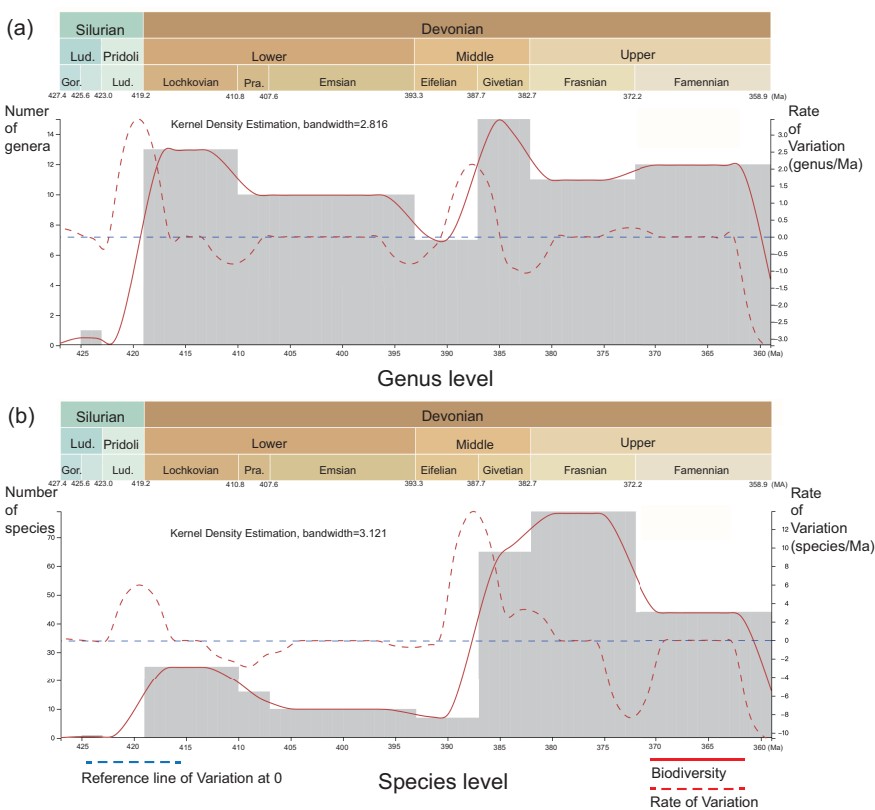


**Figure 6 Biodiversity curves of Antiarcha. A, biodiversity curve at genus level. B, biodiversity curve at species level. Histograms indicate the diversity of Antiarcha in each time interval. The continuous red line indicates the biodiversity. The dash red line indicates the rate of variation. The dash blue line is the reference line of Variation at 0. Both red lines are calculated by Kernel Density Estimation.**

**5 Data Availability**

The current dataset achived via Zenodo represents a static version of the databset in 3 November 2021:https://doi.org/10.5281/zenodo.5639529 (Pan and Zhu, 2021). The latest version of the dataset is always freely available via https://deepbone.org/ (last access: 3 November 2021).

**6 Conclusions**

Data are significant for quantitative analysis and contribute to data-driven scientific research. Previous works of early vertebrates usually focused on anatomy and phylogeny because of their significant impact on the origin of key characters. Nowadays, new collaborations have focused on biostratigraphy and biogeography, which throw new lights on the study of early vertebrates. Even if the fossil record is

incomplete, it is well demonstrated that the biogeographic evolution pattern of antiarchs can be

summarized from the compiled data and new analysis methods.

**Author contributions.** MZ supervised the project. ZHP and ZBN developed the model and performed the simulations. ZHP prepared and revised the manuscript with contributions from MZ and ZBN. ZHP and ZMX prepared the data.


**Competing interests.** The authors declare that they have no conflict of interest.

**Acknowledgement.** We thank Wenjin Zhao, Zhikun Gai, and Tuo Qiao, Institute of Vertebrate Paleontology and Paleoanthropology (CAS); Junxuan Fan, School of Earth Sciences and Engineering,

Nanjing University; Honghe Xu, Nanjing Institute of Geology and Palaeontology, Chinese Academy of Sciences (CAS); Junqi Wu, Yaqi Xue, College of Intelligence and Computing, Tianjin University for discussion and help.

**Financial support.** This project was supported by the Strategic Priority Research Program of the Chinese

Academy of Sciences (XDA19050102, XDB26000000), National Science Foundation of China (No. 42002015), Youth Innovation Promotion Association CAS (No. 2021070), State Key Laboratory of Palaeobiology and Stratigraphy (Nanjing Institute of Geology and Palaeontology, CAS) (No. 193121).

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
