# Peer review of "Novel Mid-Paleozoic dataset of antiarch placoderms (the most basal jawed vertebrates)"

_Earth System Science Data, 2021_

## Referee Comment (RC2)

[referee-annotated manuscript omitted]

---

## Community Comment (CC3)

[revised manuscript text omitted]

The main objective of this study is to present an unprecedented structured dataset of Antiarcha that potentially facilitates the research of the geospatial distribution patterns for antiarchs and the quantitive study on early vertebrates. This dataset is online open-sharing, which means that it follows the FAIR (findability, accessibility, interoperability, and reusability) guiding principles for scientific data management and stewardship. FAIR will not only maximize the usage of data but also keep it updated with new publications. In contrast to the unstructured data, this standardized dataset is much easier to access by tools and interpret by analysis. 
[revised manuscript text omitted]
. Paleontologists, stratigraphers, and evolutionary biologists could import this tab-delimited file for various purposes, such as biodiversity analysis, stratigraphic correlation, and molecular clock calibration. With the information about 6025 specimens, our dataset is far more comprehensive than the other sources. Data are significant for quantitative analysis and contribute to data-driven scientific research. Previous works of early vertebrates usually focused on anatomy and phylogeny because of their significant impact on the origin and evolution of key characters. Nowadays, new collaborations have focused on biostratigraphy and biogeography, which might throw new light on the study of early vertebrates. Hence, urgent action is required to create a trusted database on early vertebrates.

[revised manuscript text omitted]

---

## Author Response (AR1)

**Response to Referees' comments:**

**General:**

We sincerely thank two referees for their careful reading, helpful comments, and constructive suggestions, which have significantly improved the manuscript. We have carefully considered all reviewer comments and made the changes to the manuscript requested accordingly. Below we provide the point-by-point responses. Our response is given in normal font, and changes/additions to the manuscript are given in the blue text in the supplement. We hope the manuscript after careful revisions meet your high standards. The authors welcome further constructive comments if any.

**Point-to-point response to the referees:**

**Referee #1**

*The title, 'Dataset of antiarch placoderms (the most basal jawed vertebrates) throughout Middle Paleozoic', why Middle palaeozoic? In the text, we know the geological range of placodems is from the late Silurian to the Late Devonian. why not limit the time range before the fossil group? Authors need to give explanation. In the description of the dataset, the geological background is very important but not well given in the text.*

**Response**: "Middle Paleozoic" is a conventional term for Silurian and Devonian, frequently used in the literature. The international geoscience programme also uses this term, like IGCP 491 "Middle Palaeozoic Vertebrate Biogeography, Palaeogeography and Climate", and IGCP 596 " Mid-Paleozoic climate change and biodiversity patterns". For a short title, we keep it unchanged. In the Introduction section, we add "throughout Middle Paleozoic" in the sentence "Antiarcha was a diverse and successful group within Placodermi from the late Silurian to the end of Devonian…"

We have added extra text to describe the geological background in the

Introduction and 3.1 Data Overview section.

*This dataset was extracted from the DeepBone database, or the present dataset is a subset of it. The purpose of this study, as a data description study, is to show the dataset and its potential using, not giving much attention to the analytical result. The explanation of the data elements, its geological background, and data preparation, make up the key contents of the study, which, in this respect, the study should give more information. The using and analyzing data, in this study, are actually only examples.*

**Response:** The present dataset was derived from a subset of the DeepBone database. We are trying to compile all the published data of Vertebrate Paleontology one group by one group based on the DeepBone database. Dataset of Antiarcha is the first finished group. However, this dataset differs from DeepBone in the data fields. We deleted the original dataset's uninformative fields and added the paleo-coordinates fields for the practical aim.

We have strengthened the description by explaining the data elements, geological background, and data preparation.

*The valuable feature of the present dataset is its unique and abundant records of Silurian to Devonian Antiarcha. A simple comparison is given in this study (section 3.1). But I think that authors can go further.*

**Response:** Adopted. We compared the two datasets in more detail in section 3.1.

*Fossil occurrence-based dataset is better for analyzing fossil organism diversity and distribution. A lot of paleobiological study just prefer fossil occurrence data. GBDB is geological section based (Xu et al., 2020, ESSD. the publication year is 2020, but in this study it was written as 2021) and better in stratum correlation, but its data can be exported to fossil*

*occurrences.*

**Response:** Corrected.

*For the present data analyzing examples, I see that authors are still using the fossil occurrence data (figure 5 and related text). What is the unique merit of the fossil specimen-based dataset? Why the present dataset or DeepBone chose the specimen-based data structure?*

**Response:** The fossil occurrence data is suitable for data visualization. DeepBone chose the specimen-based data structure for three reasons. Firstly, because classic paleontology is based on the specimen, all the information about the specimen could be digitized with as much as possible. Secondly, a specimen with literature is the hard evidence in Paleontology. Thirdly, specimen-based data structure could cover the fossil occurrence-based data structure and do more than occurrence-based data like fossil abundance analysis, geometric morphometrics, fossil calibrations, and so on.

*Additionally, the elements in the table 1 are not all corresponding to those in the first line in the data spreadsheet.*

**Response:** Revised.

*Line 16 and other, "The dataset consists of 64 genera and 6025 records, covering all antiarch lineages". Why authors do not mention the number of species? Such thing occurs in all the text. Here "6025 records', I guess, means 6025 pieces of fossil specimens. I think such causes confusing because that it needs further definition, especially to define the basic unit (element) of the dataset.*

**Response:** This is a historical problem on the *Bothriolepis* and *Asterolepis*, the largest two groups of Antiarcha. Identifying a specimen depends on the ability to recognize species in a way that is coherent within a particular genus and through the broader groups. This is very difficult to fossil material by two

especially intractable problems: practically, by the fragmentary nature of the fossil, and philosophically by questions with the criteria by which on demarcates fossil species ( Nelson, 1999; Thomson and Thomas, 2001). For example, Thomson and Thomas (2001) reviewed the previous study on *Bothriolepis* proposed that *B. nitida, B. minor, B. virginiensis, B. Darbiensis,* and *B. colocadensis* could not be consistently distinguished. Weems (2004) questioned the validity of *B. virginiensis*. No consensus on the species level of *Bothriolepis* and *Asterolepis*. Thus, the former researchers only used the evidence of Antiarcha on genus level to discuss the biostratigraphic significance (Lelievre and Goujet, 1986; Pan, 1981; Young et al., 2010; Young and Lu, 2020).

Here '6025 records' are 5867 fossil specimens and 158 virtual specimens. Virtual specimens are introduced to store the taxon information when no precise specimen was referenced in the literature.

*In the sections 2.4 and 2.5, figures 3-5, what are the Antiarcha records? Are they individual species, localities? Or specimens? It is only obvious that basic element of the diversity analyses is the fossil taxa (figure 6).*

**Response:** Revised. In sections 2.4 and figures 3-5, they are individual specimens. In section 2.5, biodiversity is calculated on the number of genera and species. We revised the description to ensure clarity.

*Line 19 and other, "data of Antiarcha", "structured data of…: what does this mean? What data? here also need definition.*

**Response:** Revised. Data of antiarch is the information on antiarchs, which is usually the unstructured data in the literature text. Structured data is extracted from the text into a predefined format. We rewrote the description in the abstract.

*Line 21, "including testing hypotheses", actually, using data is not 'testing'*

*something but showing something.*

**Response:** We were trying to introduce the potential implementation of structured data of Antiarcha. For example, we could calculate the similarity among different areas to test the 'stepping-stone' hypotheses on the dispersal of antiarchs proposed by Li et al. (1993). Because this is a data description paper, we delete this part to meet the scope of ESSD.

*Section 1, authors should emphasize the significant of the present dataset, not only the fossil group. Such two points are closely related but different.*

**Response:** Revised.

*Lines 49-50, "Explaining the spatial and temporal distribution of early vertebrates is the prerequisite to understand their biogeographic exchange". The normal sequence is, collecting data – analyzing and showing the distribution – recognizing pattern, the last step is probably the explaining you called here.*

**Response:** Revised.

*Line 116 and others, the TrackPoint V 7.0, I only searched this software in the method part of Xu et al., 2020. Palaeogeography, Palaeoclimatology, Palaeoecology. 560. 110029.*

**Response:** The TrackPoint V 7.0 was first introduced by Ke et al. (2016) in the caption of Fig.4 on page 11. Christopher R. Scotese developed it based on modern geographical coordinates of brachiopod localities.

*Figure 5 needs to be improved, currently it is not clear and hard to get information.*

**Response:** Revised. We adjusted the contrast and saturation of the images. And we added more description in the figure caption.

*Line 221, "Based on our dataset, the oldest record of", are you sure that using dataset can conclude the time range result of a fossil? The section 4.1 seems not quite related to the present study. please reconsider it.*

**Response:** The time range of a kind of fossil is based on the stratigraphic horizons of their specimens. Paleontologists, stratigraphers, and systematic biologists are always interested in the earliest fossil record because it can be applied to stratigraphic correlation and molecular clock dating.

*Line 256, Eem event, needs explanation.*

**Response:** Revised.

*Section 6, specific and definite conclusion is needed.*

**Response:** Revised.

**Reference:**

Ke, Y., Shen, S. Z.., Shi, G. R., Fan, J. X., Zhang, H., Qiao, L., and Zeng, Y.: Global brachiopod palaeobiogeographical evolution from Changhsingian (Late Permian) to Rhaetian (Late Triassic), Paleogeogr. Paleoclimatol. Paleoecol., 448, 4-25, doi:10.1016/j.palaeo.2015.09.049, 2016.

Lekuevre, H. and Goujet, D., Biostratigraphic significance of some uppermost Devonian palcoderms, Annales de la Société géologique de Belgique, edited by Ministry of Economic Affairs, Belgian Geological Survey, pp. 55-59, 1986.

Pan, K.: Devonian antiarch biostratigraphy of China, Geol. Mag., 118(1): 69-75, 1981.

Li, Z. X., Powell, C. McA., and Trench, A.: Palaeozoic global reconstructions, in: Palaeozoic Vertebrate Biostratigraphy and Biogeography, edited by: Long, J. A., pp. 25–53, Belhaven Press, London, 1993.

Nelson, J. S.: Fishes of the World, 2nd, Jonh Wiley & Sons, New York, 523 PP, 1984.

Thomson, K. S. and Thomas, B.: On the status of species of *Bothriolepis* (Placodermi, Antiarchi) in North America, J. Vertebr. Paleontol., 21(4), 679-686, doi: 10.1671/0272-4634(2001)021[0679:OTSOSO]2.0.CO;2, 2001.

Weems, R. E.: *Bothriolepis virginiensis*, a valid species of placoderm fish separable from *Bothriolepis nitida*, J. Vertebr. Paleontol., 24(1): 245-250, 2004.

Young, G. C., Burrow, C. J., Long, J. A., Turner, S., and Choo, B.: Devonian macrovertebrate assemblages and biogeography of East Gondwana (Australasia, Antarctica), Palaeoworld, 19, 55-74, doi: 10.1016/j.palwor.2009.11.005, 2010.

Young, G. C. and Lu, J.: Asia-Gondwana connections indicated by Devonian fishes from Australia: palaeogeographic considerations, J. Palaegeogr., 9(8), 1-22, doi:10.1186/s42501-020-00057-x, 2020.

**Referee #2**

*Most of my specific comments are embedded in the annotated PDF file, and I urge the authors to go through these. Most of my remarks are related to language issues, and less on the scientific content presented in this manuscript. I believe it would be helpful if the authors received the assistance of a professional text and language editor.*

**Response**: Adopted. We revised the manuscript following the annotated PDF file accordingly. The comments which we did not follow are explained below.

Line 17: The main point of this paper is to introduce the Antiarcha dataset within the DeepBone database instead of the DeepBone database.

Line 24: Giving the data storage address at the end of the abstract is the style of ESSD referring to the other publications on its website.

Line 26: 'grade' for a paraphyletic group, and 'clade' for a monophyletic group. As such, 'grade' is correct here. To be neutral, we now replace 'grade' with 'group'.

Line 35: 'successful vertebrate group' has been used by Long (2011) 'Though gone today, placoderms ruled the planet for nearly 70 million years, making them the most successful vertebrate group of their time.' We have added the citation.

Line 37: In the following sentence, we gave examples illustrating how Antiarcha fossils contribute to the Devonian stratigraphic correlation.

Line 204: 'East Gondwana' and 'South China' are terms of biozonation following the previous study of early vertebrates. See Zhao and Zhu, 2010; Young and Lu, 2020.

*Point 1. While the prime objective of this study is clearly focused on presenting the database, the authors do however generate diversity curves and discuss the results. I would strongly recommend that the authors complement their richness assessment by computing sample-corrected curves via rarefaction, SQS, or other available method. I would also suggest that the authors expand the method section and provide detail information about the various analytical steps. The current version of the manuscript lacks sufficient detail to understand the main results. Additionally, the use of 'rate of variation' within the context of their richness assessment is vague and should be elaborated on. Finally, the inclusion of the kernel density estimator equation does not add anything here, and it would have been better if you made use of citations were appropriate.*

**Response**: Adopted. We generate the richness assessment by the divDyn R package to obtain sample-corrected curves, see figure 7. Compared to the

subsampling approaches, the old version of the diversity curve is inadequate. So, we deleted parts involving the old diversity curve, such as 'rate of variation' and 'kernel density estimate'. We agree that the subsampling approach is suitable for ameliorating some fossil-record bias.

*Point 2. As a rule of thumb, it is always better to separate results (i.e., literal reading of the data) from the discussion (i.e., the interpretation). This will make the text much easier to follow.*

**Response**: Adopted.

*Point 3. I think the authors should consider including a rationale for creating a new paleontological database. For instance what makes the DeepBone Database unique in comparison to the Paleobiology Database?*

**Response:** Adopted. Following the two referees' suggestions, we have revised the comparison between the two datasets in more detail in section 3.1.

*Point 4. The conclusion section does not really bring together or synthesis the core findings of this study. Consider re-stating the main objective of this study and how the data and results relate to it.*

**Response:** Adopted. We rewrite the conclusion to summarize the dataset. Because any interpretation of data is outside the scope of the regular article of ESSD, we try to avoid interpreting the findings.

*I really like figure 5, but the occurrence points could be made bigger with the use of transparency to deal with the issue of overlapping.*

**Response:** Revised. Following the two referees' suggestions, we try some ways to enhance figure 5. Finally, we adjusted the contrast and saturation of the images to achieve a better result. Moreover, we added more descriptions in the figure caption.

**Reference:**

Long, J. A.: Dawn of the Deed. Sci. Am., 304(1), 34-39, 2011.

Zhao, W.J., Zhu, M.: Siluro-Devonian vertebrate biostratigraphy and biogeography of China. Palaeoworld, 19, 4-26, 2010.

Young, G.C., Lu, J.: Asia–Gondwana connections indicated by Devonian fishes from Australia: palaeogeographic considerations. J. Palaeogeog., 9, 1-22, 2020.

---

## Author Response (AR2)

Dear Elger,

We sincerely thank your suggestions and the referees' review. In the present revision, we have carefully removed all the discussions where we show the interpretation of the dataset. We revised section 3.2 by deleting the interpretation part, only keeping the description of data visualization. Along with the content revising, we removed the figure of the biodiversity curve and modified the order of figures to keep fluency. We also add the reason for excluding species level from data visualization in Method section 2.5. We hope the major revision focusing on describing the dataset is in the scope of ESSD.

We welcome further comments, if any. We appreciate your consideration of our manuscript.

Sincerely,

Zhaohui Pan

Please kindly send all written correspondence to Min Zhu.

Min Zhu

Institute of Vertebrate Paleontology and Paleoanthropology, Chinese Academy of Sciences, P.O. Box 643, Beijing 100044, China.

E-mail: zhumin@ivpp.ac.cn

---

## Author Response (AR3)

Dear Elger,

We submit hereby a revised manuscript entitled "A Novel specimen-based Mid-Paleozoic dataset of antiarch placoderms (the most basal jawed vertebrates)" (ESSD-2021-394). We have carefully revised the manuscript text based on the referee's suggestions and given a point-by-point response. Our response is given in normal font, and changes/additions to the manuscript are given in the track-changes text in the supplement.

We hope the manuscript after careful revisions meet your high standards. The authors welcome further constructive comments, if any.

Yours sincerely,
Zhaohui Pan

Please kindly send all written correspondence to Min Zhu.
Min Zhu
Institute of Vertebrate Paleontology and Paleoanthropology, Chinese Academy of Sciences, P.O. Box 643, Beijing 100044, China.
E-mail: zhumin@ivpp.ac.cn

**Point-by-point response**
Thanks for the kind comment and revision suggestions for the manuscript. We revised the manuscript carefully.

Section 2.5 is named as 'Reason for choosing genus level to perform visualization'. This is an explanation of using data and a part of data analysis process. Please rename this title or move to related part.
**Response**: Accepted. We moved this part to **2.3 Data Processing and Quality Control.**

Lines 42-43, 179, 233, and others, there are wrong characters in genus and specie names. Probable errors of the PDF file. Please check and correct them.
**Response**: Lines 42-43, "*Bothriolepis*, *Asterolepis*, and *Pambulaspis*" is correct.
Line 179 "'*Jiangxilepus*', 'Bothriolepiodei' and '*Pterichthys*'" is the incorrect spell in PBDB. We gave the correct spells in the following sentence. E.g., *Jiangxilepis* (Jia et al., 2010)*, '*Bothriolepidoidei' (Miles, 1968), and *Pterichthyodes* (Hemmings and Rostron, 1972).
Line 233, "*Liujiangolepis suni*" (Wang, 1987) is correct.
We carefully checked all the genus and species names and corrected the spelling mistake in other parts (see track-changes).

Line 59, a reference on the FAIR principle is need to give.
**Response**: Adopted.

Line 203, 'equator' should be 'paleo-equator'
**Response**: Adopted.

Line 218. Two suggestions about the figure 6, 1) give numerous ages to each small stage; 2) some colors, e.g. pink and red, are not easy to discern when plotting closeron the map. Please consider and make revision.
**Response**: Revised. We added the numerous ages to each stage and increased the contrast between the two colors

**Reference:**
Hemmings, S. K., and Rostron, J.: A multivariate analysis of measurements on the Scottish Middle Old Red Sandstone antiarch fish genus *Pterichthyodes* Bleeker. Biological Journal of the Linnean Society, 4(1), 15-28, 1972.
Jia, L. T., Zhu, M., and Zhao, W. J.: A new antiarch fish from the Upper Devonian Zhongning Formation of Ningxia, China. Palaeoworld, 19(1-2), 136-145, 2010.
Miles, R. S.: The old red sandstone antiarchs of Scotland: Family bothriolepididae. Monographs of the Palaeontographical Society, 122(522), 2-127, 1968.
Wang, S. T.: A new antiarch from the Early Devonian of Guangxi, Vertebr. Palasiat., 25, 81-90, 1987.